# *Candida albicans PPG1*, a serine/threonine phosphatase, plays a vital role in central carbon metabolisms under filament-inducing conditions: A multi-omics approach

Mohammad Tahseen A. L. Bataineh[1,2,3,4]*, Nelson Cruz Soares[2,5], Mohammad Harb Semreen[2,5], Stefano Cacciatore[6,7], Nihar Ranjan Dash[1], Mohamad Hamad[2,8], Muath Khairi Mousa[5], Jasmin Shafarin Abdul Salam[5], Mutaz F. Al Gharaibeh[1], Luiz F. Zerbini[6], Mawieh Hamad[2,8]*

**1** College of Medicine, University of Sharjah, Sharjah, UAE, **2** Research Institute for Medical & Health Sciences at University of Sharjah, Sharjah, UAE, **3** Center for Biotechnology, Khalifa University of Science and Technology, Abu Dhabi, UAE, **4** Department of Genetics and Molecular Biology, College Of Medicine And Health Sciences, Khalifa University of Science and Technology, Abu Dhabi, UAE, **5** Department of Medicinal Chemistry, College of Pharmacy, University of Sharjah, Sharjah, UAE, **6** Cancer Genomics Group, International Centre for Genetic Engineering and Biotechnology, Cape Town, South Africa, **7** Institute for Reproductive and Developmental Biology, Imperial College, London, United Kingdom, **8** Department of Medical Laboratory Sciences, College of Health Sciences, University of Sharjah, Sharjah, UAE

* mohammad.bataineh@ku.ac.ae (MTALB); mabdelhaq@sharjah.ac.ae (MH)

## Abstract

*Candida albicans* is the leading cause of life-threatening bloodstream candidiasis, especially among immunocompromised patients. The reversible morphological transition from yeast to hyphal filaments in response to host environmental cues facilitates *C. albicans* tissue invasion, immune evasion, and dissemination. Hence, it is widely considered that filamentation represents one of the major virulence properties in *C. albicans*. We have previously characterized Ppg1, a PP2A-type protein phosphatase that controls filament extension and virulence in *C. albicans*. This study conducted RNA sequencing analysis of samples obtained from *C. albicans* wild type and *ppg1Δ/Δ* strains grown under filament-inducing conditions. Overall, *ppg1Δ/Δ* strain showed 1448 upregulated and 710 downregulated genes, representing approximately one-third of the entire annotated *C. albicans* genome. Transcriptomic analysis identified significant downregulation of well-characterized genes linked to filamentation and virulence, such as *ALS3*, *HWP1*, *ECE1*, and *RBT1*. Expression analysis showed *that essential* genes involved in *C. albicans* central carbon metabolisms, including *GDH3*, *GPD1*, *GPD2*, *RHR2*, *INO1*, *AAH1*, and *MET14* were among the top upregulated genes. Subsequent metabolomics analysis of *C. albicans ppg1Δ/Δ* strain revealed a negative enrichment of metabolites with carboxylic acid substituents and a positive enrichment of metabolites with pyranose substituents. Altogether, Ppg1 *in vitro* analysis revealed a link between metabolites substituents and filament formation controlled by a phosphatase to regulate morphogenesis and virulence.

**Data Availability Statement:** All relevant data are within the manuscript and its Supporting Information files.

**Funding:** MTA/1701090226-P, MH/1901050144, University of Sharjah, Sharjah, UAE. the Research Institute for Medical and Health Sciences, University of Sharjah UAE. This work was supported by research grants. The funders had no role in study design, data collection and analysis, decision to publish, or preparation of the manuscript.

**Competing interests:** The authors have declared that no competing interests exist.

**Abbreviations:** DEG, Differentially Expressed Genes; G.O., Gene Ontology; FDR, False Discovery Rate; PCA, Principal Component Analysis; W.T., Wild-Type; G.C.–M.S., Gas chromatography-mass spectrometric.

## Introduction

*Candida albicans* is the most common opportunistic fungal pathogen in humans. It causes mucosal and systemic infections and is of medical significance due to its ability to cause hospital-acquired bloodstream infections with high mortality [1,2]. *C. albicans* is part of the normal flora of the oral cavity and the gastrointestinal and genitourinary tracts in healthy individuals. Immunocompromised hosts, including cancer patients on chemotherapy, organ transplant recipients, and patients with indwelling catheters, often develop disseminated candidiasis, a systemic form of the disease with close to 40% mortality [3–5]. Antifungal therapies available to treat systemic candidiasis are limited, and current therapies have adverse side effects [6,7].

*C. albicans* is a successful commensal with no known reservoirs outside the mammalian host and possesses multiple virulence properties that cause disease. A significant virulence property is the ability to undergo a dimorphic shift from single oval-shaped cells (yeast) into elongated cells attached end-to-end (pseudohyphal and hyphal filaments) in response to host environmental conditions [8–10]. Hyphal filaments are associated with virulence and virulence-related properties, including tissue invasion, lysis of macrophages and neutrophils, and breaching of endothelial cells [11–13]. Numerous genes expressed during the reversible morphogenic switch encode proteins that play crucial roles in virulence, such as secreted aspartic proteases (SAPs), which facilitate tissue damage, and adhesins, important for attachment of *C. albicans* to host surfaces [14,15]. Furthermore, transition to the hyphal form endows *C. albicans* with the ability to evade innate immunity [16]. Various host environmental stimuli induce *C. albicans* yeast-hyphal transition through a coordinated expression of transcriptional regulators influencing multiple signal transduction pathways, including the MAP kinase pathway and the Ras-cAMP-Protein Kinase A (PKA) pathway, among others [17]. During morphological switching, the activity of signaling pathway components is controlled by kinases and fine-tuned by phosphatases [18]. For example, *C. albicans* can survive harsh environmental conditions within the host owing to their ability to produce rapid and robust stress responses. Stress-activated protein kinase (SAPK) pathways tightly regulate these stress responses [19]. However, the contribution of phosphatases in fungal stress responses remains ambiguous.

Metabolic adaptation is vital for dimorphic switching in pathogenic yeast. During yeast to hyphal morphologic transition, *C. albicans* cells grown under hyphae-inducing conditions showed an overall downregulation of cellular metabolism and significant downregulation of carbon metabolism metabolic pathways [20]. Similarly, another metabolomic study performed with azole sensitive and resistant *C. albicans* strains identified a significant change in metabolic processes such as amino acid metabolism, tricarboxylic acid cycle, and phospholipid metabolism during the development of resistance to azole drugs [21]. Previous studies have shown that *Candida* strains deficient for *PPG1*, which encodes a putative serine/threonine PP2A phosphatase, remain locked in yeast or short germ tube forms and show reduced virulence mouse model of systemic candidiasis [22]. PP2A phosphatases help maintain cell wall integrity, actin cytoskeleton organization, auxin signaling in plant cells, polar movement, and mitophagy [23–26]. Although *PPG1* was previously reported to plays a vital role in *C. albicans* filament extension and virulence, its exact role in the multitude of transcriptional networks and signaling pathways that regulate fungal morphogenesis remains unclear. To gain more insight into the poorly understood role of phosphatases in controlling *C. albicans* filamentation and virulence, we performed a sequence-based analysis of RNA samples obtained from wild-type and *ppg1Δ/Δ C. albicans* cells grown under filament-inducing conditions as means of defining *PPG1*-related transcriptomic signatures. To complement our transcriptomics findings, we performed a detailed analysis of the metabolomic profiles of wild-type and *ppg1Δ/Δ C.*

*albicans* to understand further the bearing of *PPG1* on crucial metabolic pathways under different growth conditions.

## Materials and methods

### Strains, media, and culture conditions

Wild-type *C. albicans* (DK318) and *ppg1Δ/Δ* (MAY34) strains were used throughout this study as described previously [27]. Yeast extract-peptone-dextrose (YEPD) medium at 30˚C was used as the standard non-filament-inducing growth condition for all strains. Liquid serum and temperature induction experiments were performed by growing strains overnight in YEPD medium at 30˚C to an optical density at 600 nm ($OD_{600}$) of ∼4.0 and diluting 1:10 into 50 ml of pre-warmed YEPD medium plus 10% serum at 37˚C as described previously [27]. Aliquots of cells harvested at specific post-induction time points 3 and 5 hours for RNA isolation. We selected 3 and 5 hours-time points as these times points show the most prominent phenotypic (filamentation) difference between *C. albicans* wild-type and *ppg1Δ/Δ* strains as shown before [28].

### RNA isolation, purification, and sequencing

RNA extraction was performed using RNeasy Micro Kit (Qiagen GmbH, Germany), following the manufacturer's instructions. Three biological replicates were obtained for each condition (W.T. and mutant). RNA sequencing was performed at BGI Group, Shenzhen, China. RNA was extracted from 8 samples belonging to two different strains, DK318 and MAY34 are described in Table 1.

### RNA sequencing and filtering

RNA obtained from 8 samples (Table 1) was sequenced using the BGISEQ-500 platform (Shenzhen, China), generating on average about 23.93 million reads per sample. The average mapping ratio with reference genomes was 96.92%, and the average gene mapping ratio was 82.77%; a total of 6,113 genes were detected. Sequence reads containing low-quality, adaptor-polluted and/or unknown high base (N) content were excluded from any further analysis. Retained sequence reads were further filtered using internal software SOAPnuke to produce a set of "clean reads" that was stored in FASTQ format as per each sample. Composition filtering statistics of raw data and quality metrics of clean reads are shown in S1 Table in S1 File.

### Gene mapping analysis

Clean filtered sequence reads were mapped to the reference genome using HISAT (hierarchical indexing for spliced alignment of transcripts) to perform the genome mapping step. Reads

**Table 1. Data description.**

| Sample ID | Description | Strain |
|---|---|---|
| PC3 | *ppg1Δ/Δ* strain at 3 hrs. under 30˚C control | MAY34 |
| PC5 | *ppg1Δ/Δ* strain at 5 hrs. under 30˚C control | MAY34 |
| PS3 | *ppg1Δ/Δ* strain at 3 hrs. under 37˚C + serum | MAY34 |
| PS5 | *ppg1Δ/Δ* strain at 5 hrs. under 37˚C + serum | MAY34 |
| WC3 | Wild-type strain at 3 hrs. under 30˚C control | DK318 |
| WC5 | Wild-type strain at 5 hrs. under 30˚C control | DK318 |
| WS3 | Wild-type strain at 3 hrs. under 37˚C +serum | DK318 |
| WS5 | Wild-type strain at 5 hrs. under 37˚C +serum | DK318 |

were mapped to the *C. albicans* strain SC5314 reference genome (assembly 21) (http://www.candidagenome.org). On average, 96.92% reads were mapped, and the uniformity of the mapping result for each sample suggested that the samples were comparable.

## Gene expression analysis

Clean reads were mapped to reference transcripts using Bowtie2 [29] v2.2.5, and gene expression levels in each sample were calculated using RSEM [30] v1.2.12; mapping details are shown in S2 Table in S1 File.

## Metabolite extraction and derivatization

The exact number of cells was used for each sample to avoid the effect of variable cell numbers. A volume of 300μL of the extraction solvent (acetonitrile: water, 1:1 v/v) was added to the cell pellets (2 million cells per pellet). The cells were then vortexed for 2 min to ensure the quantitative extraction of the metabolites and then stored in ice for one h, during which the samples were vortexed every 15 min. The insoluble cell matrices were then centrifuged (13,000 rpm, 10 min, −4°C).

The supernatants were collected and transferred to G.C. vials for drying using EZ-2 Plus (GeneVac-Ipswich, UK) at 37 ± 1°C. Polar metabolites such as amino acids and saccharides cannot be analyzed directly by G.C. due to their low volatility. Hence, it was necessary to derivatize them before the G.C.–M.S. analysis. The dry samples were dissolved in 25μL of 20 mg/mL methoxyamine hydrochloride in pyridine, followed by vortexing for 2 min, and stored for at least six h at 25°C before the silylation step. Next, 25 μL of N-Methyl-N-(trimethylsilyl) trifluoroacetamide (MSTFA) + 1% Trimethylchlorosilane (TMCS) were added, which were then dissolved in 100 μL of pyridine and vortexed for 2 min. For complete derivatization, the samples were incubated at 50°C for 30 min and then transferred to 200μL micro-inserts and analyzed by G.C.–M.S.

## Gas chromatography-mass spectrometric analysis of the samples

G.C.–M.S. analysis was performed using a QP2010 gas chromatography-mass spectrometer (GC-2010 coupled with a G.C.–MS QP-2010 Ultra) equipped with an auto-sampler (AOC-20i +s) from Shimadzu (Tokyo, Japan), using Rtx-5ms column (30 m length × 0.25 mm inner diameter × 0.25 μm film thickness; Restek, Bellefonte, PA, USA). Helium (99.9% purity) was used as the carrier gas with a 1 mL/min column flow rate. The column temperature regime was initially adjusted at 35°C for 2 min, followed by an increase in a rate of 10°C/min to reach 250°C. The temperature was then increased by 20°C/min until reaching 320°C and kept for 23 min. The injection volume and injection temperatures were 1 μL and 250°C using splitless injection mode, respectively. The mass spectrometer operated in electron impact mode with electron energy of 70 eV. The ion source temperature and the interface temperature were set at 240°C and 250°C, respectively. The MS mode was set on scan mode starting from 50–650 m/z with a scan speed of 1428. Data collection and analysis were performed using MSD Enhanced Chemstation software (Shimadzu). G.C. total ion chromatograms (TIC) and fragmentation patterns of the compound were identified using the NIST/EPA/NIH Mass Spectral Library (NIST 14) (S1 File). The run time for each sample was 43.67 min [31].

## Preprocessing gas chromatography-mass spectrometry (GC-MS) data

Preprocessing of metabolomics data was performed using an in-house R script. Probabilistic Quotient Normalization [32] normalizes data due to dilution effects in the extraction

procedure using the function normalization in the R package KODAMA [33]. The number of missing metabolites in the three replicates of each condition (i.e., drug and cell line) was counted. When the number was equal to 3, missing values were imputed with zero; otherwise, missing values were imputed using the k-nearest neighbor (kNN) algorithm [34], with k = 2. By limiting the kNN imputation to the metabolites with at least two values out of 3/condition, imputation using the information from different conditions (e.g., treated and non-treated) was avoided.

### Data analysis and statistical approach

DEGseq [35] and PossionDis [36] algorithms were used to detect the differentially expressed genes (DEG) between samples and groups. Hierarchical clustering for DEGs was performed using the R function heatmap. Mfuzz [37] v2.34.0 was used to cluster gene expression data for time series. Overrepresentation analysis (ORA) was used to determine the gene ontology (G. O.) functional enrichment of DEGs using the hypergeometric test using the R function phyper. Then we calculate the false discovery rate (FDR) for each p-value. In general, it was defined as significantly enriched in terms where FDR was not larger than 0.01. Principal component analysis (PCA) was used to visualize the metabolomic data. Data were mean-centered and scaled to unit variance before PCA. Metabolite set enrichment analysis (MSEA) was carried out using the Gene Set Enrichment Analysis (GSEA) algorithm [38]. The metabolite sets were built using the substituents characterization provided by the Human Metabolome Database [39]. The ranking in the MSEA was performed by using the coefficient of the first principal component of PCA. Two-way ANOVA was used to compare the strains, treatment and to investigate their interaction. The threshold for significance was $p < 0.05$ to account for multiple testing, a false discovery rate (FDR) was calculated using the q conversion algorithm in multiple comparisons. For metabolomics analysis, an FDR of $<5\%$ was chosen to reduce the identification of false positives. All data, including the raw QGD files, has been deposited to Metabolomics Workbench (https://www.metabolomicsworkbench.org). The data track ID is 2026.

## Results

### Transcriptomic changes in the C. albicans ppg1Δ/Δ strain growing under strong filament-inducing conditions

The pattern of gene expression in wild-type (DK318) and *ppg1Δ/Δ* (MAY34) strains of *C. albicans* growing under filament-inducing (10% fetal bovine serum at 37˚C) at different time points post culture (3 and 5 hours) was investigated as means of tracking any transcriptional changes that occur during the morphological transition from yeast to hyphae. The filamentation phenotype of both wild-type (W.T.) and *ppg1Δ/Δ C. albicans* strains under filamentation induction conditions was confirmed microscopically and was consistent with previously published observations [28]. As expected, all cells grew as yeast at the 37˚C non-inducing control condition, just before induction and their gene expression profile shows comparable results between *ppg1Δ/Δ* and W.T. strains. A Scatter plot of DEGs analyses of the RNA sequencing data was used to visualize the transcriptomic changes. The *ppg1Δ/Δ* strain relative to W.T. (W. S. *vs.* P.S.) at 5 hours post-induction with serum at 37˚C and showed a significant effect for *ppg1Δ/Δ* mutation on *C. albicans* gene expression at the 5-hour time point, genes that showed >2-fold change in expression levels were considered as differentially expressed. A total of 2,158 genes showed a significant difference in expression between WS5 and PS5 (Fig 1A), more detailed information can be found at the S1 File. Based on these parameters, 1448 DEGs were upregulated, and 710 were downregulated in the *ppg1Δ/Δ* samples relative to W.T.

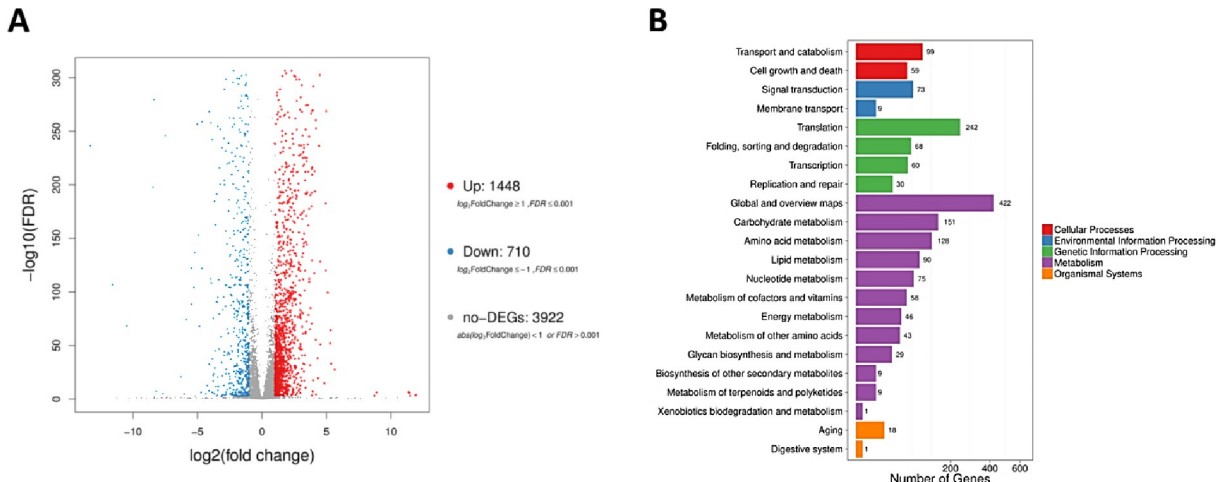

**Fig 1. Transcriptomic changes of *C. albicans ppg1Δ/Δ* strain growing under strong filament-inducing condition, 10% serum at 37˚C.** (A) Scatter plots of DEGs, X and Y axes represent $log_{10}$ transformed gene expression level, red color represents significantly upregulated genes, and blue color represents significantly down-regulated genes in W.S. and P.S. at the 5 hr. time point. (B) Pathway classification of DEGs. The X-axis represents the number of DEG. Y-axis represents the functional classification of KEGG. There are seven KEGG pathways: Cellular Processes, Environmental Information Processing, Genetic Information Processing, Metabolism, Organismal Systems.

control at the 5 hr. time point. We further classified the identified DEGs according to the proposed functional pathway, 1061 DEGs identified under metabolism KEGG pathways (Fig 1B).

*PPG1* modulates the expression of multiple filament-specific and central carbon metabolisms genes in response to serum at 37˚C: To identify essential gene targets affected by *PPG1*, we examined genes with a four-fold or more change expression between W.S. and P.S. (Table 2).

This list of genes included filament-specific genes such as *ALS3*, *HWP1*, *ECE1*, *RBT1*, and genes involved in *C. albicans* central carbon metabolisms such as *GDH3*, *GPD1*, *GPD2*, *RHR2*, *INO1*, *AAH1*, and *MET14*.

We further explored top enriched G.O. terms from DEGs based on various metabolic processes and signaling pathways. To further explore the effect of *PPG1* on various biological

**Table 2. List of selected gene targets with a four-fold or more change in expression in *ppg1 Δ/Δ* strain.**

| Gene Symbol | log2 Fold Change | P-Value | FDR | Gene Function |
|---|---|---|---|---|
| GDH3 | 7.77957 | 0.000 | 0.000 | oxidoreductase activity, acting on the CH-NH2 group of donors, NAD or NADP as acceptor glutamate dehydrogenase (NADP+) activity |
| INO1 | 7.202084 | 0.000 | 0.000 | inositol-3-phosphate synthase activity phospholipid biosynthetic process |
| AAH1 | 5.312624 | 1.14E-65 | 5.10E-65 | metal ion binding adenine deaminase activity |
| GPD2 | 4.991099 | 0.000 | 0.000 | NAD binding protein homodimerization activity |
| MET14 | 4.75566 | 0.000 | 0.000 | ATP binding adenylylsulfate kinase activity |
| GPD1 | 4.688903 | 0.000 | 0.000 | NAD binding protein homodimerization activity |
| RHR2 | 4.653127 | 0.000 | 0.000 | hydrolase activity glycerol-3-phosphatase activity |
| JEN2 | -13.4639 | 0.000 | 0.000 | dicarboxylic acid transmembrane transporter activity transmembrane transporter activity |
| HWP1 | -5.2734 | 1.03E-105 | 6.22E-105 | hyphal cell wall adhesion molecule binding |
| ECE1 | -4.84915 | 3.54E-69 | 1.66E-69 | hypha-specific protein with toxin activity |
| ALS3 | -4.80271 | 1.16E-19 | 2.81E-19 | agglutinin-like sequence adhesins |
| RBT1 | -4.452919 | 1.49E-19 | 1.97E-06 | cell wall protein |

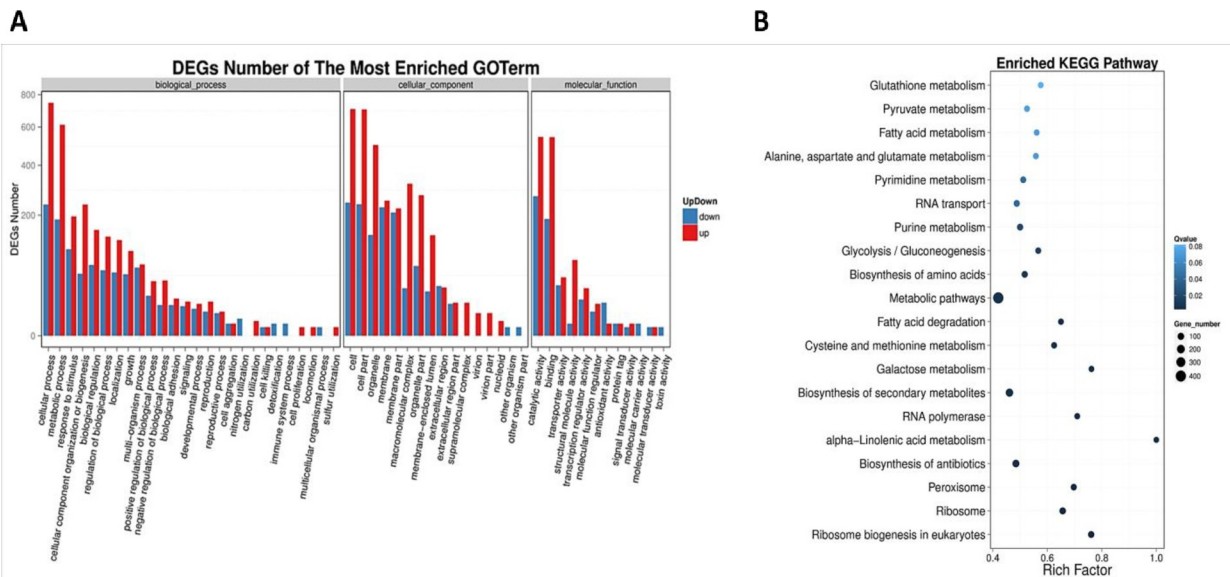

**Fig 2. Global transcriptomic analysis and pathway functional enrichment of DEGs in *C. albicans ppg1* Δ/Δ strain growing under strong filament-inducing condition.** (A) Bar graph representation of significantly up/down-regulated genes, X-axis, represents G.O. term. Y-axis represents the amount of retained G.O. terms for biological processes in RNA sequencing analysis in response to filament-induction. (B) The X-axis represents the enrichment factor. Y-axis represents the pathway name. Rich Factor refers to the value of enrichment factor, the quotient of foreground value (the number of DEGs), and background value (total gene amount). The larger the value, the more significant enrichment.

functions, we determined the functional pathway enrichment of DEGs ([Fig 2A]). In order to detect which metabolic pathways were affected by *PPG1*, we conducted a Kyoto Encyclopedia of Genes and Genomes (KEGG) analysis. This analysis detected a total of 20 enriched KEGG pathways that were significantly downregulated in *C. albicans ppg1*Δ/Δ strain growing under filament-inducing conditions ([Fig 2B]). The data showed significant enrichment of multiple metabolic pathways, such as those involved in sugar (galactose) (P-value 0.0003, FDR 3.965E-03), amino acid (P-value 0.0009, FDR 8.7187E-03), nucleotide (purines) (P-value 0.0023, FDR 1.987E-02), and fatty acid (alpha-linolenic acid) metabolism (P-value 9.357E-05, FDR 2.795E-03) among others ([Fig 2B]).

**Metabolomics analysis of C. albicans ppg1Δ/Δ strain growing under strong filament-inducing.** Using GC-MS, we profiled 35 metabolites (15 sugars and 20 amino acids). First, we examined the variance in the metabolic profiles among different samples using PCA ([Fig 3A]).

The first component (PC1) accounted for 34.5% of the total variance in the data set, with a further 22.2% explained by the second component (PC2). In the first component, we noted that the serum treatment had a considerable effect on the metabolic profile of the W.T. strains. However, this effect was not observable in the treatment of the *ppg1*Δ/Δ strain. Then, we performed enrichment analysis on the metabolites that contributed to determining the first principal component scores using their substituent characterization. According to the pre-ranked MSEA based on the first principal component's coefficient, we observed a negative enrichment of metabolites with carboxylic acid substituents (p-value = $5.75 \times 10^{-3}$; FDR = $1.32 \times 10^{-2}$) and a positive enrichment of metabolites with pyranose substituents (p-value = $5.60 \times 10^{-2}$; FDR = $4.47 \times 10^{-1}$). More detailed results of the 26 substituents metabolite sets are provided in [Table 3]. Further, we did not detect a significant difference between both strains at non-inducing control conditions.

Among the metabolites included in the carboxylic acid substituent group, we observed a statistically significant interaction between genotype (*ppg1*Δ/Δ, W.T. strains) and treatment

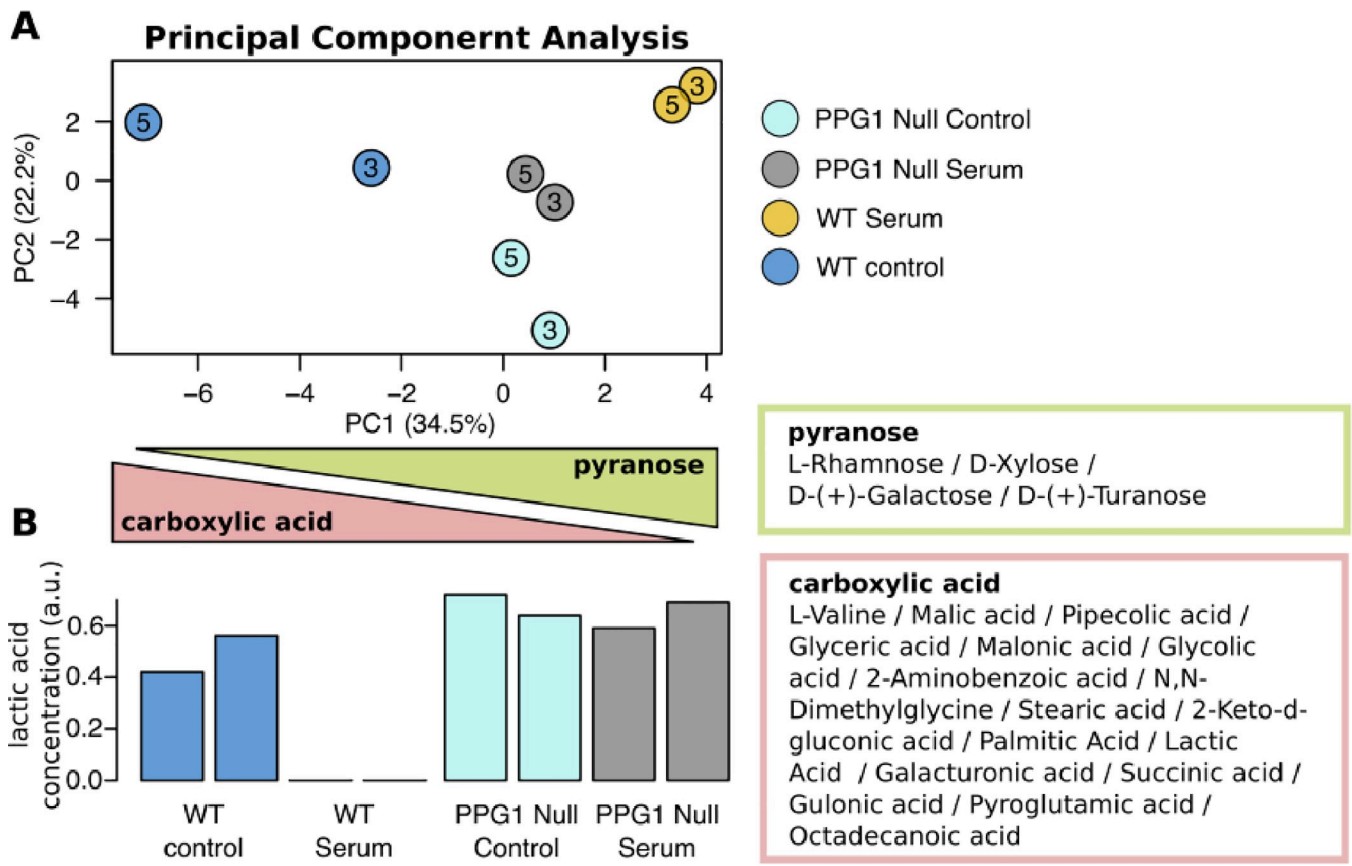

**Fig 3.** (A) PCA of the metabolomics profile in *C. albicans* under filament-inducing (10% fetal bovine serum at 37°C) and control for wild-type *C. albicans* (DK318) and *ppg1Δ/Δ* (MAY 34) strains. Cells were harvested at three and five hours. (B) The relative concentration of lactic acid. PC1 = first principal component; PC2 = second principal component.

with (serum at 37°C) on the lactic acid levels (p-value = $1.77 \times 10^{-2}$; FDR = $1.27 \times 10^{-1}$) using ANOVA. Indeed, we observed a complete depletion of lactic acid only in the W.T. strain when treated with serum (Fig 3B). ANOVA analysis on all metabolites is reported in S3 Table in S1 File.

## Discussion

Phosphorylation is an essential post-translational modification step that is highly conserved across all eukaryotes' signaling events. In *Candida*, kinases drive most cellular biologic functions, including metabolism, filamentation, and virulence [40]. Phosphatase (dephosphorylation) counters incessant kinase activity and maintains cellular homeostasis in response to different environmental stimuli [41]. This study provides new insight into the contribution of phosphatases in *C. albicans* morphogenesis utilizing transcriptomic and metabolomics approaches. Ppg1, a serine/threonine protein phosphatase, plays a vital role in controlling *C. albicans* morphology and virulence [28]. To further explore the poorly understood role of phosphateses in *C albicans* morphology and virulence, we carried out detailed transcriptomic and metabolomics profiling of wild-type and ppg1 mutants strains of the pathogen. The data showed that *C. albicans* ppg1 Δ/Δ strain growing under strong filament-inducing conditions undergo significant transcriptomic changes. The hierarchical clustering and scatter plot of DEGs showed changes in >35% of the entire *Candida* genome, 1448 upregulated genes, and

**Table 3. List of metabolites associated with Ppg1.**

| pathway | p-value | FDR | ES | NES | size | metabolites |
|---|---|---|---|---|---|---|
| Carboxylic acid | 5.75E-03 | 1.32E-01 | -0.75 | -1.77 | 17 | L-Valine/Malic acid/Pipecolic acid/Glyceric acid/Malonic acid/Glycolic acid/ 2-Aminobenzoic acid/N,N-Dimethylglycine/Stearic acid/2-Keto-d-gluconic acid/Palmitic Acid/Lactic Acid/Galacturonic acid/Succinic acid/Gulonic acid/Pyroglutamic acid/ Octadecanoic acid |
| Pyranose | 5.60E-02 | 4.47E-01 | 0.85 | 1.74 | 4 | L-Rhamnose/D-Xylose/D-(+)-Galactose/D-(+)-Turanose |
| Beta-hydroxy acid | 6.61E-02 | 4.47E-01 | -0.80 | -1.41 | 4 | Malic acid/Glyceric acid/Galacturonic acid/Gulonic acid |
| Polyol | 8.16E-02 | 4.47E-01 | 0.45 | 1.39 | 12 | Pentitol/Glycerol/2-Keto-d-gluconic acid/L-Rhamnose/Galacturonic acid/D-Xylose/D-(+)-Galactose/Gulonic acid/Scyllo-Inositol/D-(+)-Turanose/Xylitol/Myo-Inositol |
| Oxacycle | 1.17E-01 | 4.47E-01 | 0.60 | 1.38 | 5 | Uridine/L-Rhamnose/D-Xylose/D-(+)-Galactose/D-(+)-Turanose |
| Amino acid | 1.18E-01 | 4.47E-01 | -0.74 | -1.31 | 4 | L-Valine/Pipecolic acid/2-Aminobenzoic acid/N,N-Dimethylglycine |
| Dicarboxylic acid or derivatives | 1.36E-01 | 4.47E-01 | -0.80 | -1.29 | 3 | Malic acid/Malonic acid/Succinic acid |
| Fatty acid | 1.81E-01 | 5.20E-01 | -0.61 | -1.25 | 7 | L-Valine/Malic acid/Galacturonic acid/Succinic acid/Gulonic acid/Palmitic Acid/ Octadecanoic acid |
| Carbonyl group | 2.34E-01 | 5.40E-01 | -0.51 | -1.21 | 19 | L-Valine/Malic acid/Pipecolic acid/Glyceric acid/Malonic acid/Glycolic acid/N, N-Dimethylglycine/Stearic acid/2-Keto-d-gluconic acid/Palmitic Acid/Lactic Acid/Glycerol monostearate/Galacturonic acid/Succinic acid/Gulonic acid/Pyroglutamic acid/Oleic acid amide/D-(+)-Turanose/Octadecanoic acid |
| Alpha-hydroxy acid | 2.35E-01 | 5.40E-01 | -0.61 | -1.20 | 6 | Malic acid/Glyceric acid/Glycolic acid/Lactic Acid/Galacturonic acid/Gulonic acid |
| Carboxylic acid derivative | 3.12E-01 | 6.27E-01 | -0.49 | -1.13 | 13 | Malic acid/Glyceric acid/Glycolic acid/2-Aminobenzoic acid/Stearic acid/2-Keto-d-gluconic acid/Palmitic Acid/Lactic Acid/Glycerol monostearate/Galacturonic acid/Gulonic acid/Oleic acid amide/Octadecanoic acid |
| Fatty acyl | 3.48E-01 | 6.27E-01 | -0.62 | -1.09 | 4 | L-Valine/Glycerol monostearate/Galacturonic acid/Gulonic acid |
| Sugar alcohol | 3.54E-01 | 6.27E-01 | 0.45 | 1.03 | 5 | Pentitol/Glycerol/Scyllo-Inositol/Xylitol/Myo-Inositol |
| Alcohol | 4.41E-01 | 7.25E-01 | -0.43 | -1.03 | 20 | Ergosterol/Octadecanol/Pentitol/Malic acid/Glyceric acid/Glycolic acid/Glycerol/2-Keto-d-gluconic acid/Lactic Acid/Glycerol monostearate/Uridine/Acetoin/Tryptophol/L-Rhamnose/Galacturonic acid/D-Xylose/D-(+)-Galactose/Gulonic acid/D-(+)-Turanose/Xylitol |
| Organonitrogen compound | 5.52E-01 | 8.47E-01 | -0.46 | -0.96 | 8 | L-Valine/Pipecolic acid/2-Aminobenzoic acid/N,N-Dimethylglycine/Uridine/Tryptophol/ Pyroglutamic acid/Oleic acid amide |
| Primary alcohol | 6.69E-01 | 8.98E-01 | -0.37 | -0.86 | 13 | Octadecanol/Pentitol/Glyceric acid/Glycolic acid/Glycerol/2-Keto-d-gluconic acid/Glycerol monostearate/Uridine/Tryptophol/D-(+)-Galactose/Gulonic acid/D-(+)-Turanose/Xylitol |
| Monosaccharide | 6.87E-01 | 8.98E-01 | -0.44 | -0.86 | 6 | Pentitol/Glycerol/Glyceric acid/Uridine/Galacturonic acid/Gulonic acid/Xylitol |
| Cyclic alcohol | 7.14E-01 | 8.98E-01 | -0.53 | -0.85 | 3 | Ergosterol/Scyllo-Inositol/Myo-Inositol |
| Azacycle | 7.55E-01 | 8.98E-01 | -0.47 | -0.83 | 4 | Pipecolic acid/Uridine/Tryptophol/Pyroglutamic acid |
| Ketone | 8.13E-01 | 8.98E-01 | 0.41 | 0.74 | 3 | 2-Keto-d-gluconic acid/Acetoin/D-(+)-Turanose |
| Secondary alcohol | 8.20E-01 | 8.98E-01 | -0.30 | -0.72 | 17 | Ergosterol/Pentitol/Malic acid/Glyceric acid/Glycerol/2-Keto-d-gluconic acid/Lactic Acid/ Uridine/2,3-Butanediol/Acetoin/L-Rhamnose/Galacturonic acid/D-Xylose/D-(+)-Galactose/ Gulonic acid/D-(+)-Turanose/Xylitol |
| Aliphatic heteromonocyclic compound | 8.79E-01 | 9.19E-01 | -0.34 | -0.67 | 6 | Pipecolic acid/L-Rhamnose/D-Xylose/D-(+)-Galactose/Pyroglutamic acid/D-(+)-Turanose |
| Organoheterocyclic compound | 9.82E-01 | 9.82E-01 | -0.26 | -0.53 | 7 | Pipecolic acid/Uridine/L-Rhamnose/D-Xylose/D-(+)-Galactose/Pyroglutamic acid/D-(+)-Turanose |

Abbreviations: ES = enrichment score; NES = normalized enrichment score; FDR = false discovery rate.

710 downregulated genes compared to the W.T. control. Consistent with previous findings, this significant transcriptional change suggests an important, and possibly a master regulator, the role of *PPG1* in filament extension, among other potential roles [28,41,42]. *PPG1* role is not surprising given that ser/thr phosphatases consist of multiprotein complexes with significant structural diversity that provides for an expansive array of regulatory roles in multiple signaling events.

 

Among the critical targets for the Ppp1 regulatory effect were filament-specific and central carbon metabolism pathways. Consistent with the previous analysis, the most downregulated genes in *ppg1 Δ/Δ C. albicans* grown at 37˚C were genes involved in filamentation and virulence, such as *ALS3*, *HWP1*, *ECE1*, and *RBT1* [43]. It is well accepted that *C. albicans* cells utilize Als3 (a member of the agglutinin-like sequence adhesins) during filamentation for epithelial adhesion [44]. Moreover, Als3 plays a significant role in iron acquisition, which is critical for fungal pathogenesis [45]. Interestingly, we noticed an upregulation in the expression of the iron transport gene *FET31* in *C. albicans ppg1Δ/Δ* strain. *FET31* upregulation could reflect increased metabolism [46] in the *ppg1Δ/Δ* strain as suggested by the transcriptomic profile shown in Figs 1 and 2. Furthermore, our data suggest that *PPG1* could function as a significant regulator given that it increased the expression of multiple carbon metabolism genes, including *GDH3*, *GPD1*, *GPD2*, *RHR2*, *INO1*, *AAH1*, and *MET14. GDH3* (NADPH-dependent glutamate dehydrogenase). This group of genes is collectively essential for nitrogen metabolism, the maintenance of the redox balance, and *C. albicans* filament formation [47,48]. For example, *GPD1* and *GPD2* (two isoforms of glycerol 3-phosphate dehydrogenase) are rate-controlling enzymes in essential glycerol formation reactions in *Saccharomyces cerevisiae* [49]. They also play a crucial role in osmoregulation, carbohydrate metabolism, and redox balancing [47,50]. It is worth noting that both Gpd1 and Gpd2 are negatively regulated by the phosphorylation activity of the AMP-activated protein kinase Snf1, the TORC2-dependent kinases Ypk1 and Ypk2 possibly in a Ppg1-dependant manner [51]. Additionally, Candida glycerol 3-phosphate dehydrogenases help the pathogen evade the immune response through their ability to interact with the vital complement regulators, including H and H–like factors [52]. *RHR2* (glycerol 3-phosphatase) is also essential for osmotic stress, glycerol accumulation, biofilm formation, and yeast-hyphal switch [53,54]. *INO1* (Inositol-1-phosphate synthase), which is vital for inositol synthesis, is considered as a growth factor that supports the formation of glycophosphatidylinositol (GPI)-anchored glycolipids on *Candida* cell surface and hence the promotion of pathogenesis [55,56]. *AAH1* (an Adenine deaminase) is similar to serine/threonine dehydratases essential for purine salvage and nitrogen catabolism [47]. *MET14* (an adenylylsulfate kinase) is essential for assimilating sulfate to sulfide, which strongly depends on yeast growth conditions such as glucose [57].

Our data showed that a total of 20 enriched KEGG pathways were significantly downregulated in *C. albicans ppg1Δ/Δ* strain growing under filament-inducing conditions (Fig 2B). These mainly included pathways involved in sugars (galactose) and amino acid biosynthesis and purine metabolism, all of which are essential for filament extension and virulence [58,59]. Interestingly, the highest pathway enrichment in *C. albicans ppg1Δ/Δ* strain grown under filament growth conditions concerns the metabolism of alpha-Linolenic acid (ALA), a known inhibitor of hyphal growth *C. albicans*. Previous transcriptional profiling revealed that ALA downregulates hypha-specific genes in a *UME6* and *RFG1* (hyphal transcriptional regulators) independent manner. Perhaps ALA functions through a Ppg1- dependent mechanism [60].

Based on the above-noted discussion, we sought to explore further the effect of *PPG1* on *C. albicans* metabolic functions using GC-MS metabolomics analysis. We profiled 35 metabolites, including sugars and amino acids. Enrichment analysis showed an adverse enrichment profile of metabolites with carboxylic acid substituents in *C. albicans ppg1Δ/Δ* strain growing under strong filament-inducing conditions and consistent with the downregulation of Jen2 (a dicarboxylic acid transporter protein), which was previously shown to be regulated by glucose repression in *C. albicans* as shown in Table 1 [61]. *C. albicans* uses carboxylic acids substituents including acetate and lactate for survival and evasion of phagocytosis [62,63]. For example, macrophages-mediated phagocytosis during systemic candidiasis in mice was reported to induce lactate and acetate transporters [63]. Positive enrichment of metabolites with pyranose

substituents in *C. albicans ppg1*Δ/Δ strain growing under strong filament-inducing conditions consistent with the upregulation of genes involved in central carbon metabolism and hence fungal pathogenicity [47]. Glycans are critical for fungal pathogenesis owing to their well-established roles in cell adhesion, immune cell evasion, and inhibition of lymphoproliferation [64–66]. Mannans as a vital component of the fungal cell wall, are also of significance in fungal virulence; inactivation of genes involved in mannan biosynthesis was previously linked to decreased virulence of *C. albicans* [67]. Altogether, Ppg1 seems to exhibit a master regulator role that influences lactic acid and carboxylic acid utilization and the conversion of pyranose to sugars that could be utilized to synthesize filamentation-related cell wall polysaccharides.

In conclusion, the data presented here elaborated on the role of phosphatases such as *PPG1* in regulating the morphological transition of *C. albicans* at the transcriptional level. *PPG1* affected the expression of >35% of the *Candida* genus, especially those involved in or associated with *C. albicans* pathogenesis, filamentation, and metabolic activities. Shedding more light on the regulatory events that ensue during *C. albicans* filamentous growth and virulence may lead to novel antifungal therapeutic strategies. Further work is still needed to validate the findings presented by this study using *PPG1* mutant-induced animal models. Further analysis of the role of Ppg1 in protein glycosylation and other virulence-related events such as biofilm formation and immune evasion is also warranted.

## Supporting information

**S1 File. Supporting file contains all the supporting tables and figures, supplementary file.** S1 Fig. GC-MS total ion chromatograms (TIC) of metabolites extract from *Candida albicans*. S1 Table. Clean reads quality metrics. S2 Table. Summary of Genome Mapping Ratio. S3 Table. Metabolites interplay between *ppg1*Δ/Δ and W.T. strains. (XLSX)

## Acknowledgments

We are grateful to David Kadosh (U.T. Health, San Antonio), USA, to provide *C. albicans* strains.

## Author Contributions

**Conceptualization:** Mohammad Tahseen A. L. Bataineh, Nelson Cruz Soares, Mawieh Hamad.

**Data curation:** Mohammad Tahseen A. L. Bataineh, Mohammad Harb Semreen, Stefano Cacciatore, Muath Khairi Mousa, Jasmin Shafarin Abdul Salam.

**Formal analysis:** Mohammad Tahseen A. L. Bataineh, Nelson Cruz Soares, Mohammad Harb Semreen, Stefano Cacciatore, Nihar Ranjan Dash, Mohamad Hamad, Muath Khairi Mousa, Mawieh Hamad.

**Funding acquisition:** Mohammad Tahseen A. L. Bataineh, Mawieh Hamad.

**Investigation:** Mohammad Tahseen A. L. Bataineh, Nelson Cruz Soares, Nihar Ranjan Dash, Jasmin Shafarin Abdul Salam, Mawieh Hamad.

**Methodology:** Mohammad Tahseen A. L. Bataineh, Nelson Cruz Soares, Mohammad Harb Semreen, Nihar Ranjan Dash, Muath Khairi Mousa, Jasmin Shafarin Abdul Salam, Mawieh Hamad.

**Resources:** Mohammad Tahseen A. L. Bataineh, Nelson Cruz Soares, Mohammad Harb Semreen, Stefano Cacciatore, Muath Khairi Mousa, Luiz F. Zerbini, Mawieh Hamad.

**Software:** Nelson Cruz Soares, Mohammad Harb Semreen, Stefano Cacciatore, Mohamad Hamad, Muath Khairi Mousa, Jasmin Shafarin Abdul Salam, Mutaz F. Al Gharaibeh, Luiz F. Zerbini, Mawieh Hamad.

**Supervision:** Mohammad Tahseen A. L. Bataineh, Mohammad Harb Semreen, Nihar Ranjan Dash, Mohamad Hamad, Luiz F. Zerbini, Mawieh Hamad.

**Validation:** Mohammad Tahseen A. L. Bataineh, Nelson Cruz Soares.

**Visualization:** Stefano Cacciatore, Mutaz F. Al Gharaibeh.

**Writing – original draft:** Mohammad Tahseen A. L. Bataineh, Nelson Cruz Soares, Mohammad Harb Semreen, Stefano Cacciatore, Nihar Ranjan Dash, Mohamad Hamad, Muath Khairi Mousa, Jasmin Shafarin Abdul Salam, Mutaz F. Al Gharaibeh, Luiz F. Zerbini, Mawieh Hamad.

**Writing – review & editing:** Mohammad Tahseen A. L. Bataineh, Nelson Cruz Soares, Mohammad Harb Semreen, Stefano Cacciatore, Nihar Ranjan Dash, Mohamad Hamad, Muath Khairi Mousa, Jasmin Shafarin Abdul Salam, Mutaz F. Al Gharaibeh, Luiz F. Zerbini, Mawieh Hamad.

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
