## [Decision Letter · Decision Letter 0]

28 Jul 2021

PONE-D-21-21871

Candida albicans PPG1, a serine/threonine phosphatase, plays a vital role in central carbon metabolisms under filament-inducing conditions: a multi-omics approach.

PLOS ONE

Dear Dr. Al Bataineh,

Thank you for submitting your manuscript to PLOS ONE. After careful consideration, we feel that it has merit but does not fully meet PLOS ONE’s publication criteria as it currently stands. Therefore, we invite you to submit a revised version of the manuscript that addresses the points raised during the review process.

Two experts in this field thoroughly review this manuscript. Both of them appreciated the quantity and quality of this work, but raised a series of editorial and experimental concerns that the authors should pay attention to. In particular, the authors may need to further analyze their transcriptome and metabolomics data to obtain a better picture of PPG1-dependent signaling networks. 

We look forward to receiving your revised manuscript.

Kind regards,

Yong-Sun Bahn, Ph.D.

Academic Editor

PLOS ONE

Journal Requirements:

"The authors wish to acknowledge the generous support of the Research Institute for Medical and Health Sciences, University of Sharjah UAE. We are grateful to David Kadosh (U.T. Health, San Antonio), USA, to provide C. albicans strains. This work was supported by research grants 1701090226-P/MTA, 1901050144/M.H., University of Sharjah, Sharjah, UAE."

"MTA/1701090226-P, MH/1901050144, University of Sharjah, Sharjah, UAE.

the Research Institute for Medical and Health Sciences, University of Sharjah UAE. This work was supported by research grants. The funders had no role in study design, data collection and analysis, decision to publish, or preparation of the manuscript"

Reviewers' comments:

Reviewer's Responses to Questions

**Comments to the Author**

1. Is the manuscript technically sound, and do the data support the conclusions?

Reviewer #1: Partly

Reviewer #2: Partly

2. Has the statistical analysis been performed appropriately and rigorously? 

Reviewer #1: Yes

Reviewer #2: Yes

3. Have the authors made all data underlying the findings in their manuscript fully available?

Reviewer #1: Yes

Reviewer #2: No

4. Is the manuscript presented in an intelligible fashion and written in standard English?

Reviewer #1: Yes

Reviewer #2: Yes

5. Review Comments to the Author

Reviewer #1: This study performed combined RNA sequencing and metabolomics assay to check the detailed function of PPG1 in filamentation and virulence. Generally, this study showed plenty of data with appropriate analysis, which gives useful information. The manuscript is well written. And the author discussed the potential mechanism by which PPG1 regulates filamentation and virulence logically. But I have some suggestions and comments which may be useful for improving the manuscript.

1. The manuscript supplied enough data but didn't end with a clear story or conclusion. Actually, I still don't know how PPG1 regulate filamentation and virulence. Just saw some genes upregulated and some others downregulated. Therefore, more extended analysis is necessary to tell a logical story.

2. The author directly compares the gene expression between WT and PPG1 KO strains under the FBS inducing conditions. But to find filamentation related genes that are regulated by PPG1, it is better to know the FBS induced transcription in WT group first, and see what kind of genes are not induced or suppressed in KO group. So, I suggest the author should set the groups as WT vs WT+FBS and PPG1KO vs PPG1KO+FBS, if they have the original data.

3. The whole paper supplied enough omics information but with no confirmation. qRT PCR is necessary to confirm the transcription even they used 3 biological replicates, in view of that they neglect the P value or FDR when chosing the changed genes. In particular, the potential targets for PPG1 mentioned in discussion part need to be confirmed.

4. I believe PPG1 regulates the transcription of HWP1, ECE1, ALS3, but the high expression of HSGs, is a common effect by deleting many filamention related genes. And the author try to link the transcription data to metabolomics data, such as the author suggested ALA may function through a Ppg1- dependent mechanism. But I think it is not enough. Some bench work is needed to support the prediction, such as overexpression the targets in PPG1 KO strain and detect the metabolisms.

5. A working model of PPG1 may help to understand its role in filamentation and virulence. PPG1 itself is a serine/threonine phosphatase, a graph contains the substrate and their filamentation targets as well as metabolisms will be helpful for understanding.

Reviewer #2: Summary: The authors have previously characterized Ppg1, a PP2A-type protein phosphatase that controls filament extension and virulence in C. albicans. This study is a follow up analysis of the ppg1Δ/Δ strain in regulating transcriptome relevant to morphogenesis using RNA sequencing analysis. The authors identified that downregulation of well-characterized genes linked to filamentation and virulence as well as the genes involved in the central carbon metabolisms were down regulated in the mutant strain. Their subsequent metabolomics analysis of C. albicans ppg1Δ/Δ strain revealed a negative enrichment of metabolites with carboxylic acid substituents and a positive enrichment of metabolites with pyranose substituents. The authors concluded that Ppg1 is a link between metabolites substituents and filament formation controlled by a phosphatase to regulate morphogenesis and virulence. Overall, this manuscript is descriptive, and no functional validation was undertaken to validate the RNA-seq/metabolomics finding. The authors make a case that Ppg1 controls carbon metabolism and filamentation, but the connection to hyphae based on metabolism is not clear.

Comments:

1. The authors conducted large scale transcriptome and metabolome analysis and generated significant amount of data set. They would be able to articulate more defined biological implications of Ppg1 function with a thorough data analysis on the valuable resources they already possess. For example, it would be interesting to see how the transcriptome changes in the wild type or ppg1 mutant strains responding to 30 C control vs. 37 C serum conditions. Just for an overview, they would be able to conduct a PCA analysis (or a hierarchical clustering) of 8 samples presented in Table 1 to compare the overall transcriptome of the wild-type and the ppg1 mutant strain under different conditions.

2. The authors analyzed the transcriptome of the wild type and the ppg1 mutant strains at 3 and 5 hrs post hyphal induction. Based on their previous study published in 2014, the ppg1 mutant demonstrated a similar expression pattern of ALS3, HWP1, and ECE1 at 3 and 5 hrs. However, the expression of these genes (and NRG1) was quite distinctive at 1 or 2 hrs post-induction compared to the wild-type strain. Thus, it is expected that ppg1 mutation would have impacted the transcription of genes at the earlier stage of hyphal induction. I am wondering why the authors did not choose earlier time points for transcriptome analysis. Is there any clear rationale why the authors chose 3 and 5 hrs only? Of those two time points, the authors only used the transcriptome at 5 hrs for their data analysis and the reason was not clearly stated either.

3. Even though the authors demonstrated that Ppg1 is involved in hyphal induction, it is also possible that Ppg1 would play roles in carbon metabolism under yeast condition. Since the authors already have transcriptome data at 30 C, I would recommend comparing the transcriptome of the wild type and the mutant strains under yeast condition as well.

4. What is the control condition for the metabolomic analysis? Is it the same as the control condition for the transcriptome analysis (30C no serum)? The comparison of the control and inducing conditions were not clearly stated in the method or result sections. Please change “PPG1 null” to “ppg1 delta/delta” to be consistent with the transcriptome data.

5. Metabolite normalization: if the ppg1 strain is a slower grower, then the final cell number at the harvest time (after 3 and 5 hrs) would have been different. The presentation of the metabolites/total protein would be more appropriate for normalization.

1. In #360-374, the authors claimed that Ppg1 is involved in central carbon metabolism and cell wall architecture. However, the discussion is descriptive and has no supporting evidence. In S. cerevisiae, Ppg1 homolog is involved in glycogen accumulation. Have the authors tested glycogen accumulation in the mutant strain? If Ppg1 plays a role as a master regulator, as the authors speculated, we would expect to see altered cell wall architecture or cell wall integrity. Have the authors tested for cell wall stressor susceptibility or chitin staining to see any compensatory chitin upregulation in the mutant strain?

6. #214 at the 37C non-inducing condition – 30C?

7. #216 can you further elaborate what “comparable results” means?

8. In consistent wording for time point description: #218, 5 hours; 3219, 5-hour; #223, 5hr.

9. #237 and # 238; PPG1 (Gene), Ppg1 for protein

10. #238, “essential genes” are often used for their functional relevance to cell viability. Did you refer the genes that are directly impacted by Ppg1? Then please use a different term.

11. Table 1, could you include functional categories at the front row of this table? Carbon metabolism, hyphal specific genes, etc.

12. #266 wording in the section title: grown under filament-inducing “condition”

13. #307 “ppg1 mutants” to “ppg1 Δ/Δ”

14. #319 C. albicans to strain, please revise the sentence in #318-320.

15. #325 “increased metabolism” can you specify?

16. #336 Italicize “Candida”

17. #362 add space between “table” and “1”

18. GEO submission numbers are not found.

6. PLOS authors have the option to publish the peer review history of their article (what does this mean?). If published, this will include your full peer review and any attached files.

Reviewer #1: No

Reviewer #2: No

---

## [Author Response · Author response to Decision Letter 0]

26 Sep 2021

Academic Editor comments

Two experts in this field thoroughly review this manuscript. Both of them appreciated the quantity and quality of this work, but raised a series of editorial and experimental concerns that the authors should pay attention to. In particular, the authors may need to further analyze their transcriptome and metabolomics data to obtain a better picture of PPG1-dependent signaling networks. 

We thank the academic editor and respected reviewers for their time and effort. We have acknowledged and incorporated all reviewers' comments and modified the manuscript when possible. First, we provided a supplementary file detailing differential gene expression comparison between the eight conditions mentioned in table 1. Next, we inserted figure S4 reflecting the transcriptomic changes at 3 hrs. time point. We also modified figure 3 for the metabolomic analysis in response to reviewers' comments. Last, we modified the discussion and conclusion, highlighting the limitations of this study. 

Reviewer #1: This study performed combined RNA sequencing and metabolomics assay to check the detailed function of PPG1 in filamentation and virulence. Generally, this study showed plenty of data with appropriate analysis, which gives useful information. The manuscript is well written. And the author discussed the potential mechanism by which PPG1 regulates filamentation and virulence logically. But I have some suggestions and comments which may be useful for improving the manuscript.

1. The manuscript supplied enough data but didn't end with a clear story or conclusion. Actually, I still don't know how PPG1 regulate filamentation and virulence. Just saw some genes upregulated and some others downregulated. Therefore, more extended analysis is necessary to tell a logical story. 

We thank reviewer 1 for this suggestion. We have previously characterized PPG1 and demonstrated its importance in filamentation and virulence [1]. In particular, we showed PPG1 importance for filament-specific genes such as ALS3, HWP1, ECE1 by northern analysis. However, it was unclear how PPG1 may regulate the complex regulatory circuits that control morphology and virulence. Therefore, we conducted a global transcriptomic analysis coupled with metabolomics to understand the PPG1 role better. Consistent with previous northern plotting, the transcriptomic analysis identified significant downregulation of well-characterized genes linked to filamentation and virulence, including ALS3, HWP1, ECE1, and RBT1. Further global expression analysis showed important genes involved in C. albicans central carbon metabolisms, including GDH3, GPD1, GPD2, RHR2, INO1, AAH1, and MET14, the top upregulated genes (Table 1). Subsequent metabolomics analysis confirmed the findings from figure 2 (carboxylic acid and pyranose substituents), and then we linked these data with genes from table 1 in the discussion section, such as GDH3 # 326-337 and JEN2 #359-363. Given the eccentric nature of phosphatases and the fact that it most likely affects many other cell biology components, as shown in figure 1A, over 35% of the entire Candida genome was affected. We acknowledge the limitation and complexity of this type of analysis and agree that a more specific analysis is needed to explore different mechanisms in the future, mentioned in # 393-397. 

2. The author directly compares the gene expression between WT and PPG1 KO strains under the FBS inducing conditions. But to find filamentation related genes that are regulated by PPG1, it is better to know the FBS induced transcription in WT group first, and see what kind of genes are not induced or suppressed in KO group. So, I suggest the author should set the groups as WT vs WT+FBS and PPG1KO vs PPG1KO+FBS, if they have the original data. 

This analysis was done as a baseline control but not shown. Therefore, we added a supplementary file in # 217-220 between the different groups as requested. 

3. The whole paper supplied enough omics information but with no confirmation. qRT PCR is necessary to confirm the transcription even they used 3 biological replicates, in view of that they neglect the P value or FDR when chosing the changed genes. In particular, the potential targets for PPG1 mentioned in discussion part need to be confirmed.

In table 1 #251, we did provide the P-value and FDR of the listed genes. We agree with reviewer 1 that a confirmation of the transcriptional changes is needed. However, in this study, we determined metabolic profiles consistent with the well-characterized gene functions from table 1. While this approach may not furnish a direct relationship as acknowledged in limitation # 400-407, it provides insight at the metabolic level for future studies. 

4. I believe PPG1 regulates the transcription of HWP1, ECE1, ALS3, but the high expression of HSGs, is a common effect by deleting many filamention related genes. And the author try to link the transcription data to metabolomics data, such as the author suggested ALA may function through a Ppg1- dependent mechanism. But I think it is not enough. Some bench work is needed to support the prediction, such as overexpression the targets in PPG1 KO strain and detect the metabolisms. 

We agree with reviewer 1 and removed this prediction from discussion line # 375. 

5. A working model of PPG1 may help to understand its role in filamentation and virulence. PPG1 itself is a serine/threonine phosphatase, a graph contains the substrate and their filamentation targets as well as metabolisms will be helpful for understanding. 

We agree with reviewer 1, and we can provide a network visualization for predicted protein-protein interactions. However, a more beneficial working model can be generated for a specific mechanism of action as part of the follow-up study. 

Reviewer #2: Summary: The authors have previously characterized Ppg1, a PP2A-type protein phosphatase that controls filament extension and virulence in C. albicans. This study is a follow up analysis of the ppg1Δ/Δ strain in regulating transcriptome relevant to morphogenesis using RNA sequencing analysis. The authors identified that downregulation of well-characterized genes linked to filamentation and virulence as well as the genes involved in the central carbon metabolisms were down regulated in the mutant strain. Their subsequent metabolomics analysis of C. albicans ppg1Δ/Δ strain revealed a negative enrichment of metabolites with carboxylic acid substituents and a positive enrichment of metabolites with pyranose substituents. The authors concluded that Ppg1 is a link between metabolites substituents and filament formation controlled by a phosphatase to regulate morphogenesis and virulence. Overall, this manuscript is descriptive, and no functional validation was undertaken to validate the RNA-seq/metabolomics finding. The authors make a case that Ppg1 controls carbon metabolism and filamentation, but the connection to hyphae based on metabolism is not clear.

Comments:

1. The authors conducted large scale transcriptome and metabolome analysis and generated significant amount of data set. They would be able to articulate more defined biological implications of Ppg1 function with a thorough data analysis on the valuable resources they already possess. For example, it would be interesting to see how the transcriptome changes in the wild type or ppg1 mutant strains responding to 30 C control vs. 37 C serum conditions. Just for an overview, they would be able to conduct a PCA analysis (or a hierarchical clustering) of 8 samples presented in Table 1 to compare the overall transcriptome of the wild-type and the ppg1 mutant strain under different conditions. 

We thank reviewer 2 for all suggestions. We agree with reviewer 2 and provided supplementary file # 217-220 for the 8 conditions presented in Table 1. 

2. The authors analyzed the transcriptome of the wild type and the ppg1 mutant strains at 3 and 5 hrs post hyphal induction. Based on their previous study published in 2014, the ppg1 mutant demonstrated a similar expression pattern of ALS3, HWP1, and ECE1 at 3 and 5 hrs. However, the expression of these genes (and NRG1) was quite distinctive at 1 or 2 hrs post-induction compared to the wild-type strain. Thus, it is expected that ppg1 mutation would have impacted the transcription of genes at the earlier stage of hyphal induction. I am wondering why the authors did not choose earlier time points for transcriptome analysis. Is there any clear rationale why the authors chose 3 and 5 hrs only? Of those two time points, the authors only used the transcriptome at 5 hrs for their data analysis and the reason was not clearly stated either. 

We agree with reviewer 2 that early time points are important to explore the transcriptional kinetics and the overall effect of PPG1. However, our primary goal in this study was to explore the effect of PPG1 on filament extension, a hallmark of virulence in C. albicans. Previous literature on serum and temperature induction experiments showed a clear transition from pseudohyphae to hyphae around 3 hrs. post induction as explained in method #111-113. We also added figure S3, showing 3 hrs. time points data #229-230. 

3. Even though the authors demonstrated that Ppg1 is involved in hyphal induction, it is also possible that Ppg1 would play roles in carbon metabolism under yeast condition. Since the authors already have transcriptome data at 30 C, I would recommend comparing the transcriptome of the wild type and the mutant strains under yeast condition as well. 

We agree with reviewer 2 there are multiple combinations of comparisons that can be done, including non-inducing at 30˚C. However, we sought to highlight the most prominent ones (WT vs. mutant at 37˚C + serum at 3 and 5 hrs.). We also added supplementary file # 217-220, as mentioned before. 

4. What is the control condition for the metabolomic analysis? Is it the same as the control condition for the transcriptome analysis (30C no serum)? The comparison of the control and inducing conditions were not clearly stated in the method or result sections. Please change "PPG1 null" to "ppg1 delta/delta" to be consistent with the transcriptome data. 

We agree with reviewer 2 and have modified the statement in line # 199-202 to reflect that a "Two-way ANOVA was used to compare the strains, treatment and to investigate their interaction and analysis of the metabolite concentration was done to better understand the direction of the changes including the comparison between control and inducing factor as shown in table S3 under the column "serum treatment" and mentioned in results section # 310-311. We have also changed PPG1 null into ppg1Δ/Δ as requested in FIG.3 # 285. 

5. Metabolite normalization: if the ppg1 strain is a slower grower, then the final cell number at the harvest time (after 3 and 5 hrs) would have been different. The presentation of the metabolites/total protein would be more appropriate for normalization. 

We agree with reviewer 2 and have modified the statements in line # 179-180 "Probabilistic Quotient Normalization [32] normalizes data due to dilution effects in the extraction procedure using the function normalization in the R package KODAMA [33]. We also used Probabilist Quotient Normalization to adjust the different final cell numbers at the harvest time". However, normalizing by protein could be inappropriate since the filaments (made of proteins) could affect protein quantification. 

1. In #360-374, the authors claimed that Ppg1 is involved in central carbon metabolism and cell wall architecture. However, the discussion is descriptive and has no supporting evidence. In S. cerevisiae, Ppg1 homolog is involved in glycogen accumulation. Have the authors tested glycogen accumulation in the mutant strain? If Ppg1 plays a role as a master regulator, as the authors speculated, we would expect to see altered cell wall architecture or cell wall integrity. Have the authors tested for cell wall stressor susceptibility or chitin staining to see any compensatory chitin upregulation in the mutant strain? 

We have previously shown an important role of PPG1 in call wall adhesion, and the current study confirmed these findings with an insight to the transcriptional and metabonomic contribution toward that observation as shown before in FIG 4 [1]. that said, we agree with reviewers 2 suggestion that other cell wall and biofilm studies are warranted as mentioned # 407. 

6. #214 at the 37C non-inducing condition – 30C?

Corrected # 217. 

7. #216 can you further elaborate what "comparable results" means?

Corrected # 217-220. 

8. In consistent wording for time point description: #218, 5 hours; 3219, 5-hour; #223, 5hr.

Corrected to reflect 5 hrs. wording. 

9. #237 and # 238; PPG1 (Gene), Ppg1 for protein

Corrected # 253. 

10. #238, "essential genes" are often used for their functional relevance to cell viability. Did you refer the genes that are directly impacted by Ppg1? Then please use a different term.

Agree and corrected # 252. 

11. Table 1, could you include functional categories at the front row of this table? Carbon metabolism, hyphal specific genes, etc.

In addition to the gene function category, we can add other categories such as cellular components, biological processes, and Kegg orthology, but we do not feel this is necessary and may overload the table 1. 

12. #266 wording in the section title: grown under filament-inducing "condition"

Corrected # 283.

13. #307 "ppg1 mutants" to "ppg1 Δ/Δ"

Corrected # 325.

14. #319 C. albicans to strain, please revise the sentence in #318-320.

Corrected # 336-337.

15. #325 "increased metabolism" can you specify?

Corrected # 343-344.

16. #336 Italicize "Candida"

Corrected # 356.

17. #362 add space between "table" and "1"

Corrected # 382.

18. GEO submission numbers are not found.

Gene expression is provided in the supplimntry file, and the complete data can be provided upon acceptance. 

1. Albataineh MT, Lazzell A, Lopez-Ribot JL, Kadosh D. Ppg1, a PP2A-type protein phosphatase, controls filament extension and virulence in Candida albicans. Eukaryot Cell. 2014;13(12):1538-47. Epub 2014/10/19. doi: 10.1128/EC.00199-14. PubMed PMID: 25326520; PubMed Central PMCID: PMCPMC4248689.

---

## [Decision Letter · Decision Letter 1]

22 Oct 2021

Candida albicans PPG1, a serine/threonine phosphatase, plays a vital role in central carbon metabolisms under filament-inducing conditions: a multi-omics approach.

PONE-D-21-21871R1

Dear Dr. Al Bataineh,

We’re pleased to inform you that your manuscript has been judged scientifically suitable for publication and will be formally accepted for publication once it meets all outstanding technical requirements.

Kind regards,

Yong-Sun Bahn, Ph.D.

Academic Editor

PLOS ONE

Additional Editor Comments (optional):

Two original reviewers re-evaluated the revised manuscript, and both agreed that it was properly revised. Although the reviewer 2 made a very minor comment in Figure 3 labeling, I believe that it could be easily fixed during proofreading.

Reviewers' comments:

Reviewer's Responses to Questions

**Comments to the Author**

1. If the authors have adequately addressed your comments raised in a previous round of review and you feel that this manuscript is now acceptable for publication, you may indicate that here to bypass the “Comments to the Author” section, enter your conflict of interest statement in the “Confidential to Editor” section, and submit your "Accept" recommendation.

Reviewer #1: All comments have been addressed

Reviewer #2: All comments have been addressed

2. Is the manuscript technically sound, and do the data support the conclusions?

Reviewer #1: Yes

Reviewer #2: Yes

3. Has the statistical analysis been performed appropriately and rigorously? 

Reviewer #1: Yes

Reviewer #2: Yes

4. Have the authors made all data underlying the findings in their manuscript fully available?

Reviewer #1: Yes

Reviewer #2: Yes

5. Is the manuscript presented in an intelligible fashion and written in standard English?

Reviewer #1: Yes

Reviewer #2: Yes

6. Review Comments to the Author

Reviewer #1: The author addressed my corcerns well, unless some tables inculding Table1 and Table s3 seem incomplete.

Reviewer #2: Just a really minor comment on the Figure 3 data label; please switch the order to be WT control, WT serum, ppg1 null control and ppg1 null serum. Otherwise, the authors addressed the reviewers comments well with clarity.

7. PLOS authors have the option to publish the peer review history of their article (what does this mean?). If published, this will include your full peer review and any attached files.

Reviewer #1: No

Reviewer #2: No

---

## [Editor Report · Acceptance letter]

2 Nov 2021

PONE-D-21-21871R1 

*Candida albicans PPG1*, a serine/threonine phosphatase, plays a vital role in central carbon metabolisms under filament-inducing conditions: a multi-omics approach. 

Dear Dr. AL Bataineh:

I'm pleased to inform you that your manuscript has been deemed suitable for publication in PLOS ONE. Congratulations! Your manuscript is now with our production department. 

Kind regards, 

on behalf of

Dr. Yong-Sun Bahn 

Academic Editor

PLOS ONE